# Integration and evaluation of a gradient-based needle navigation system for percutaneous MR-guided interventions

**Li Pan**[1☯], **Steffi Valdeig**[2☯], **Urte Kägebein**[3,4], **Kun Qing**[5,6], **Barry Fetics**[7], **Amir Roth**[7], **Erez Nevo**[7], **Bennet Hensen**[3,4]*, **Clifford R. Weiss**[2], **Frank K. Wacker**[3,4]

**1** Siemens Healthineers, Baltimore, MD, United States of America, **2** Department of Radiology and Radiological Science, Johns Hopkins University, Baltimore, MD, United States of America, **3** Department of Radiology, Hannover Medical School, Hannover, Germany, **4** STIMULATE–Research Campus: Solution Centre for Image Guided Local Therapies, Magdeburg, Germany, **5** Department of Radiology and Medical Imaging, University of Virginia, Charlottesville, VA, United States of America, **6** Siemens Corporate Technology, Baltimore, MD, United States of America, **7** Robin Medical Inc., Baltimore, MD, United States of America

☯ These authors contributed equally to this work.
* hensen.bennet@mh-hannover.de

**Data Availability Statement:** All relevant data are within the manuscript and its Supporting Information files.

## Abstract

The purpose of the present study was to integrate an interactive gradient-based needle navigation system and to evaluate the feasibility and accuracy of the system for real-time MR guided needle puncture in a multi-ring phantom and in vivo in a porcine model. The gradient-based navigation system was implemented in a 1.5T MRI. An interactive multi-slice real-time sequence was modified to provide the excitation gradients used by two sets of three orthogonal pick-up coils integrated into a needle holder. Position and orientation of the needle holder were determined and the trajectory was superimposed on pre-acquired MR images. A gel phantom with embedded ring targets was used to evaluate accuracy using 3D distance from needle tip to target. Six punctures were performed in animals to evaluate feasibility, time, overall error (target to needle tip) and system error (needle tip to the guidance needle trajectory) in vivo. In the phantom experiments, the overall error was 6.2±2.9 mm (mean±SD) and 4.4±1.3 mm, respectively. In the porcine model, the setup time ranged from 176 to 204 seconds, the average needle insertion time was 96.3±40.5 seconds (min: 42 seconds; max: 154 seconds). The overall error and the system error was 8.8±7.8 mm (min: 0.8 mm; max: 20.0 mm) and 3.3±1.4 mm (min: 1.8 mm; max: 5.2 mm), respectively.

## Introduction

Image guidance for percutaneous biopsy, drainage and therapy is the current standard of care. Not only does image guidance decrease procedural risk by allowing physicians to access locations that cannot be accessed without real-time feedback, but it also improves diagnostic yield by precisely targeting focal lesions. Currently percutaneous image guided procedures rely primarily on ultrasound [1–4], computed tomography (CT) [5, 6] and X-ray fluoroscopy [7, 8].

**Funding:** This work was funded in part by the NIH/ NCI SBIR Phase II grant 2R44CA124238. Siemens Healthineers provided support in the form of salary for LP and KQ, provided grant support for KQ and CW, and provided institutional funding for FW. Robin Medical provided support in the form of salary for BF, AM, and EN. The funders did not have any role in the study design, data collection and analysis, decision to publish, or preparation of the manuscript. The specific roles of these authors are articulated in the 'author contributions' section.

**Competing interests:** The authors have read the journal's policy and the authors of this manuscript have the following competing interests: LP is a paid employee of Siemens Healthineers. KQ receives grant support from Siemens Healthineers and was an intern employee while working on this project. BF and AM are paid employees of Robin Medical. EN is the founder and a stock owner of Robin Medical, the company that developed the EndoScout tracking system. CW receives grant support from Siemens Healthineers, BTG, and Medtronic, and is a consultant for BSCI and Medtronic. FW receives institutional funding from Siemens Healthineers. There are no patents, products in development or marketed products associated with this research to declare. This does not alter our adherence to PLOS ONE policies on sharing data and materials.

Due to magnetic resonances imaging's (MRI's) high soft tissue contrast and ability to provide physiological function information, diagnostic and therapeutic procedures under MRI guidance have emerged as alternatives in recent years and have been shown to provide accurate targeting [9–13].

MRI offers unique advantages including excellent soft tissue contrast, functional as well as structural information, flexible image plane control, all without the use of ionizing radiation exposure to patient and personnel. However, the enthusiasm of radiologists to take advantage of the features of MRI to guide interventions is significantly hampered by difficult access to the patient, and lack of both interactive navigation systems and interventional user interfaces to improve workflow. With the current shorter large bore magnets access to the patient is somewhat improved. However, workflow remains problematic in interventional MRI (iMRI) and navigation tools to improve targeting would be beneficial.

Typically, navigation systems for MRI are designed to facilitate entry point and target selection, needle pathway planning, and real-time MR imaging planes definition; all of which help to target deep, small lesions and to avoid critical structure damage along the needle path. Most of the navigation systems evaluated for the use in an MRI environment offer improved targeting accuracy for needle guidance [14–19]. However, most of the navigation systems are not approved for clinical use and requires extra equipment that make them inconvenient/time-consuming to set up and use. As well, most of these systems fail to offer real-time feedback which is crucial for accurate and safe device placement [14–22].

Interactive real-time MR imaging with the capability of rapid image acquisition, interactive scan plane control and protocol parameter update, as well as automatic device tracking, has been demonstrated to be extremely useful in MR guided interventions [11, 23–25]. The gradient-based navigation system (EndoScout®, Robin Medical Inc., Baltimore, MD) offers an adjunct to interactive real-time MRI using a unique method for device tracking [26–28]. In contrast to other active tracking systems, which rely on active tracking coils that are physically connected to the MR scanner, the EndoScout system enables MR gradients to interact with the tracking coils without a physical connection. The purpose of the present work is to integrate the gradient-based tracking technique into a multi-slice interactive real-time pulse sequence, and to evaluate the accuracy of needle placement using this integrated system in both a phantom and in an animal model.

## Materials and methods

### Navigation system

The MR gradient-based EndoScout navigation system (Fig 1, EndoScout®, Robin Medical Inc., Baltimore, MD) comprises a three dimensional magnetic field cubic sensor with two sets of orthogonal pick-up micro-coils (six coils in total, Fig 2C) integrated into an instrument holder (Fig 2A). By switching gradients in X, Y and Z directions, the location and orientation of the sensor can be determined by measuring the voltages induced in the orthogonal micro-coils during gradient activations of the scanner. According to the clinical application, the sensor comes in various sizes and shapes, to suit different instruments and applications. Additionally, the needle holder can be sterilized.

The system is connected to the gradient amplifiers to detect the currents sent to the gradient coils during the gradient excitation and determine the gradient fields generated inside the magnet. The sensor is connected to the interface unit (Fig 1B) which is located inside the magnet room. The main electronic unit (Fig 1C) is located in the equipment room and receives the current output of three gradients as well as the signals from the interface units. The system then measures the voltages induced in the micro-coils during gradient transition to determine

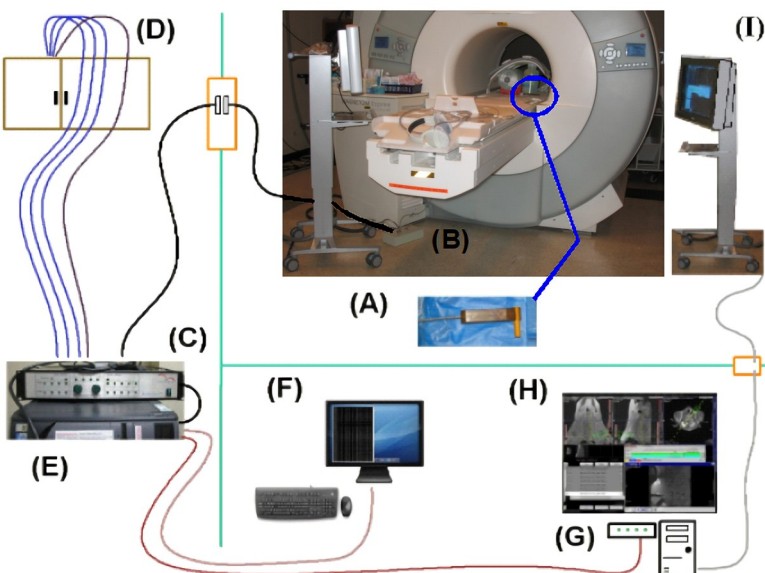

**Fig 1. Integration of the EndoScout system in the MRI environment.** The EndoScout system is configured for the Siemens Espree scanner as: signals are obtained on sensor (A) and are transmitted-to and processed-by interface unit (B) and electronic unit (C). Gradient command and trigger signals are obtained from the cabinet (D). Signals are digitized and processed by EndoScout PC (E), which is controlled remotely via KVM-extension at (F). Real-time tracking annotation overlay is transferred to scanner computer via Ethernet connection to the hub (G). Real-time tracking annotation overlay is then displayed on control-room MR console monitor (H) and in-room monitor (I).

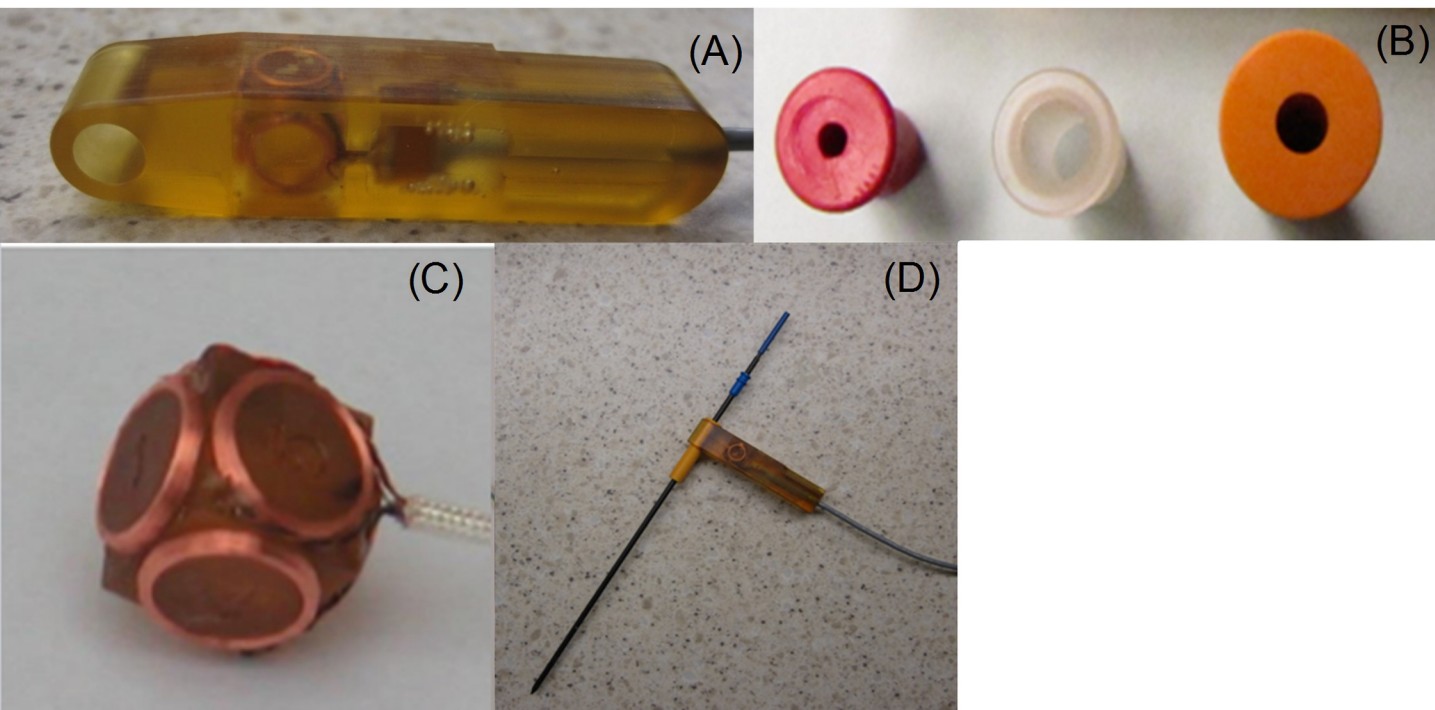

**Fig 2. The gradient-based hand-held device in the EndoScout navigation system.** (A) The needle holder (or surgical tool holder) with the MR gradient field detection sensor embedded. (B) Three different types of needle holder adapters which can attach to the needle holder and hold variable sized applicators. (C) MR gradient field detecting sensor with two sets of three orthogonal pick up coils (six coils in total). The sensor determines applicator location and orientation within the magnet, and based on the needle length, the tip position of the needle can be calculated. (D) A puncture needle is attached to the hand-held device via a needle holder adapter.

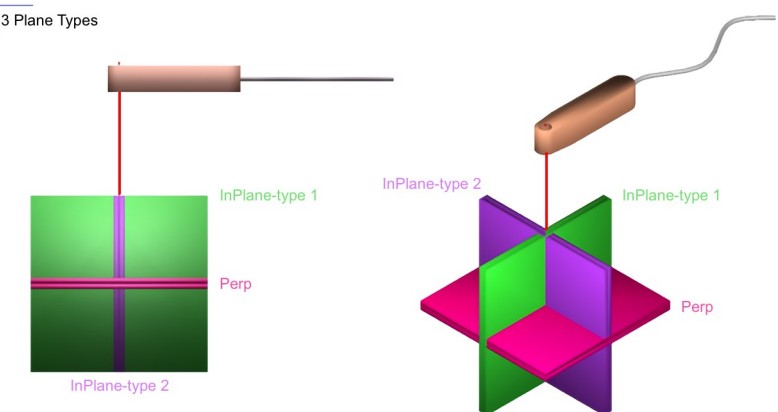

**Fig 3. Image plane definition relative to the needle holder.** Green slice: needle holder in-plane with needle tip as the center of the imaging slice. Purple slice: needle holder perpendicular to the plane with needle tip as the center of the imaging plane. Pink slice: needle (attached to the needle holder) perpendicular to the plane with needle tip as the center of the imaging plane.

the sensor position and orientation [27]. The X, Y and Z fields from gradient coils determined during system installation are used as reference.

The sensor coils are integrated into a hand-held device which is attached to a needle (Fig 2D) via needle adapter with various sizes (Fig 2B). An operator can then hold the needle holder while advancing the needle during a percutaneous procedure (Fig 2E). Based on the relative position of the sensor and the needle location, the needle orientation and the tip location can be determined.

The image plane definition relative to the needle holder is shown in Fig 3. The green and purple planes are parallel to the needle axis and would display the needle as a line projected onto the plane, and the pink plane is perpendicular to the needle axis and would display the needle as a point intersecting the plane.

### Pulse sequences

An interactive multi-slice real-time sequence (BEAT_IRTTT, Siemens Healthineers, Erlangen, Germany) with the following acquisition parameters was used: FOV 300 x 300 mm$^2$; TR, <4 ms; TE, < 3 ms 2 ms; flip angle, 50˚–70˚; acquisition matrix, 192 × 192, receiver bandwidth, > 500 Hz/pixel. This facilitated acquisition times of < 1 second per slice for 3 parallel, sequentially acquired planes 4-mm in thickness with a 1-mm gap. The sequence was modified to provide the excitation gradients and trigger signal that are fed into the EndoScout system for sensor tracking. Bipolar excitation gradients were applied in X, Y and Z directions consecutively and repeated for every phase encoding line or every slice in both TrueFISP (Fig 4) and GRE acquisitions. The repetition interval of excitation gradients determines the update interval of the sensor position and orientation. Based on the sensor tracking updates and the relative position of the sensor with respect to the surgical holder and the device, the position and orientation of the puncture needle are determined. The sequence can also automatically update the scan planes to acquire the images along the needle (containing the needle holder or perpendicular to the needle holder) or perpendicular to the needle in real-time, as defined in Fig 3.

### Experimental setup

All studies were performed in a 1.5 T system (Magnetom Espree, Siemens Healthineers, Erlangen, Germany). For procedure planning, in both phantom and animal experiments, 3D

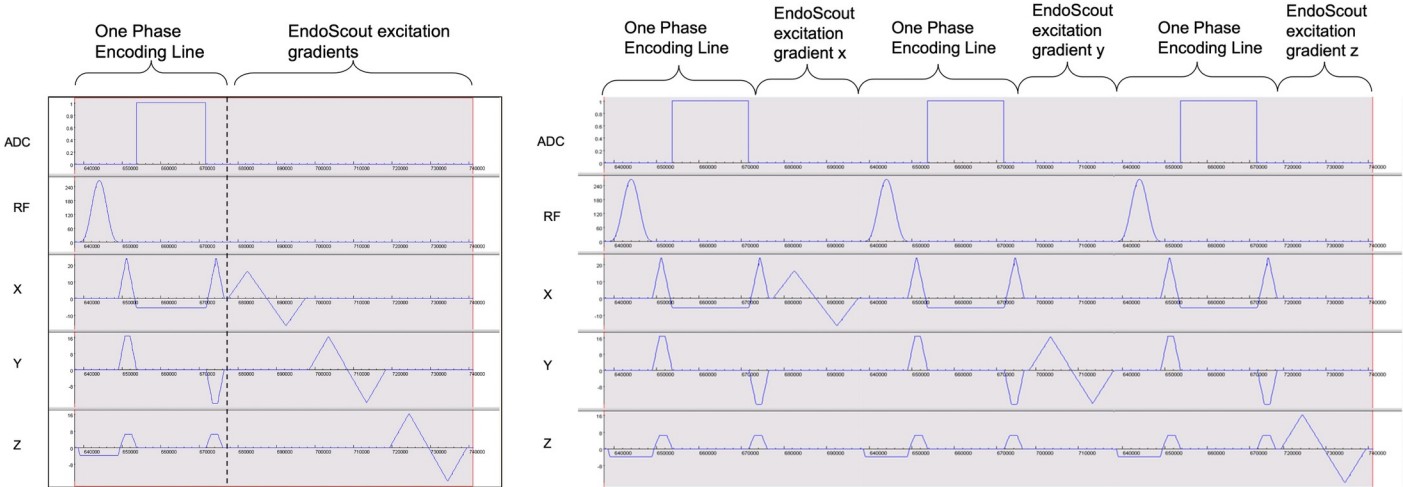

**Fig 4. Pulse sequence diagram of the integrated EndoScout system.** (A) Phase Encode Mode. The EndoScout excitation gradients applied for every phase encoding line in a TrueFISP acquisition. The three gradients in X, Y and Z directions are consecutively activated with a fixed, bi-polar pattern. It enables position update to the EndoScout system for every phase encode interval. (B) Sequential Mode. Only one gradient activation is added for each phase encode, and the three gradients are activated sequentially. This provides sufficient data for tracking for every three phase encode steps.

volume gradient echo images (TR 7.42 ms; TE 2.38 ms; res. 1.3x1.3x2.5 mm$^3$) were first acquired and the axial, coronal and sagittal reference images containing the target were identified as road-map images. In the animal experiments, additional axial, coronal and sagittal HASTE images (TR 2000 ms; TE 62 ms; res. 1.2x1.2x5.0 mm$^3$) were acquired when the target was not well visualized on the T1 weighted images. The three reference images were uploaded into three image frames on the scanner user interface and also sent to EndoScout system for registration. The entry point and the needle path were then determined with the planned needle path displayed as an intersection line on one of the roadmap images. Once the desired entry point and needle path to the target was selected, the needle was aligned using the EndoScout system.

The BEAT_IRTTT sequence with EndoScout integration using TrueFISP acquisition was used during the needle insertion for MR guidance. The additional gradient activation inserted for each phase encoding steps enables the sensor tracking with a high frame rate. During the scan, the current position and orientation of the sensor within the magnet was computed by the EndoScout system and projected needle puncture path was displayed as a graphical overlay (tracking annotation) superimposed on the roadmap MR images (Fig 5) on the scanner user interface to indicate the relative spatial relationship between the surgical device and the respective underlying three slices. The tracking annotation indicates the needle tip position (the tip of the green triangle) and the orientation of the needle shown as colored lines connecting to the triangles. The corresponding location of the needle relative to the respective slice was shown in yellow (proximal), blue (in-plane) and red (distal).

Real-time images along the needle path were acquired by manually aligning the scanning plane (using the scanner user interface) to the needle path indicated by graphical overlay. During the navigated intervention, EndoScout tracking updated the device position and orientation every 8.2 ms and the acquisition time for each slice was 377 ms (Fig 6). The tracking annotation on pre-acquired images along the planned needle path were displayed for interactive navigation as follows: three slices were displayed in the inline display window in a mosaic mode, with slice 1 always following the needle plane (containing the needle holder), and slices 2 and 3 updated with real-time images along the planned path and final target but without slice location change (Fig 7). If necessary, the operator could slightly rotate the needle holder

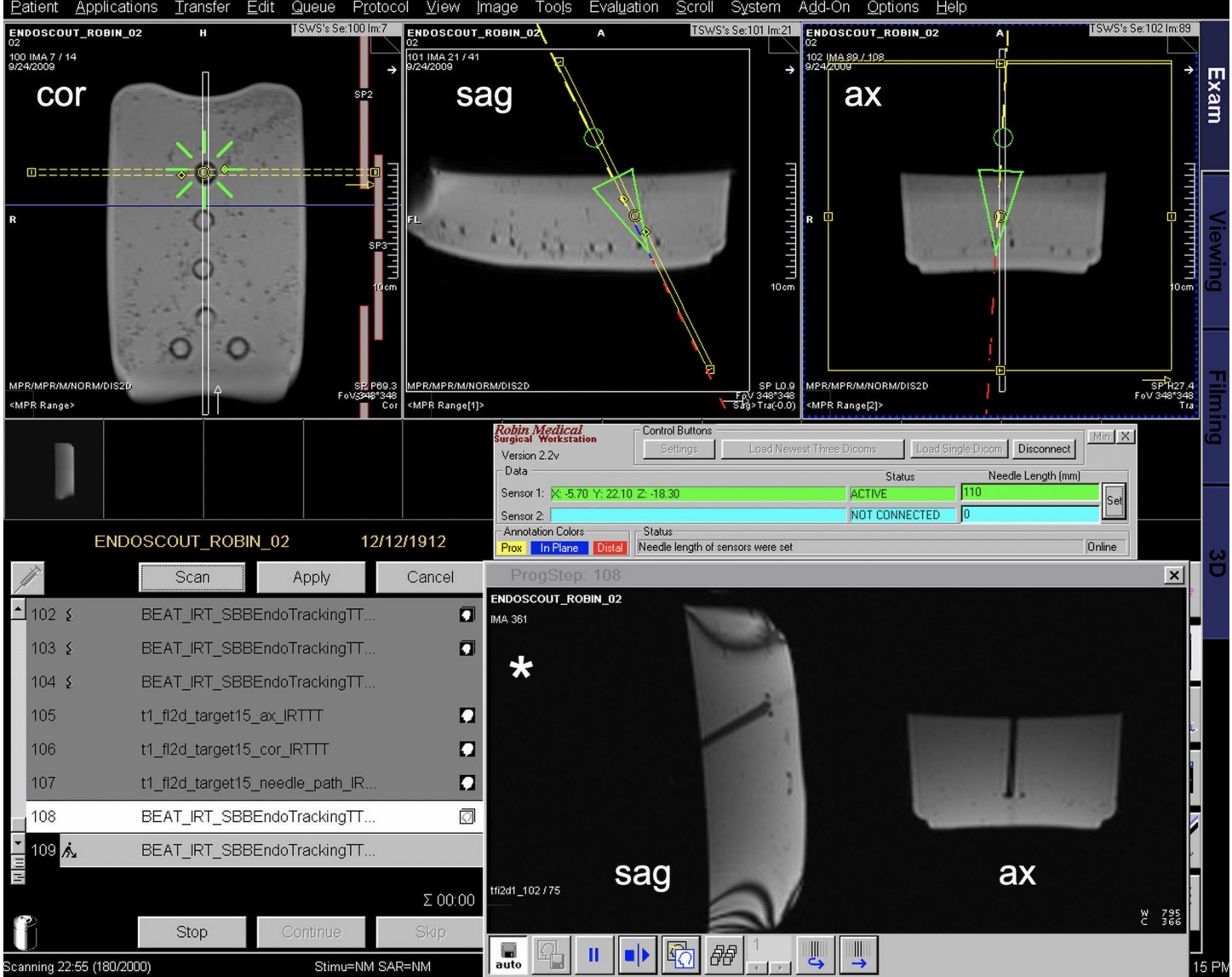

**Fig 5. Screenshot of the MR in-room monitor during a phantom experiment.** The top three windows show the overlay of the navigation system (green/yellow/red overlay) with pre-acquired MR images in coronal, sagittal, and axial orientation. On the lower right side (star) the real-time MR images with sagittal and axial views are displayed.

around the needle so that the slice orientation will be updated for better observation of surrounding tissues along the needle path. For needle guidance, on the pre-acquired axial slice, one slice containing the planned surgical path was shown as intersection plane and was used as the reference for the operator to align surgical device to keep it within the planned path (Figs 6 and 7, top right image).

A T1 GRE sequence was performed in the axial, coronal and sagittal orientation after the procedure to obtain the final needle path and the exact tip location. This information was used to check whether the needle is aligned with the planned path and to measure the overall error (Euclidian distance from the needle tip to the target) as well as the system error (needle tip to the closest point from the guidance needle trajectory calculated by the EndoScout navigation system).

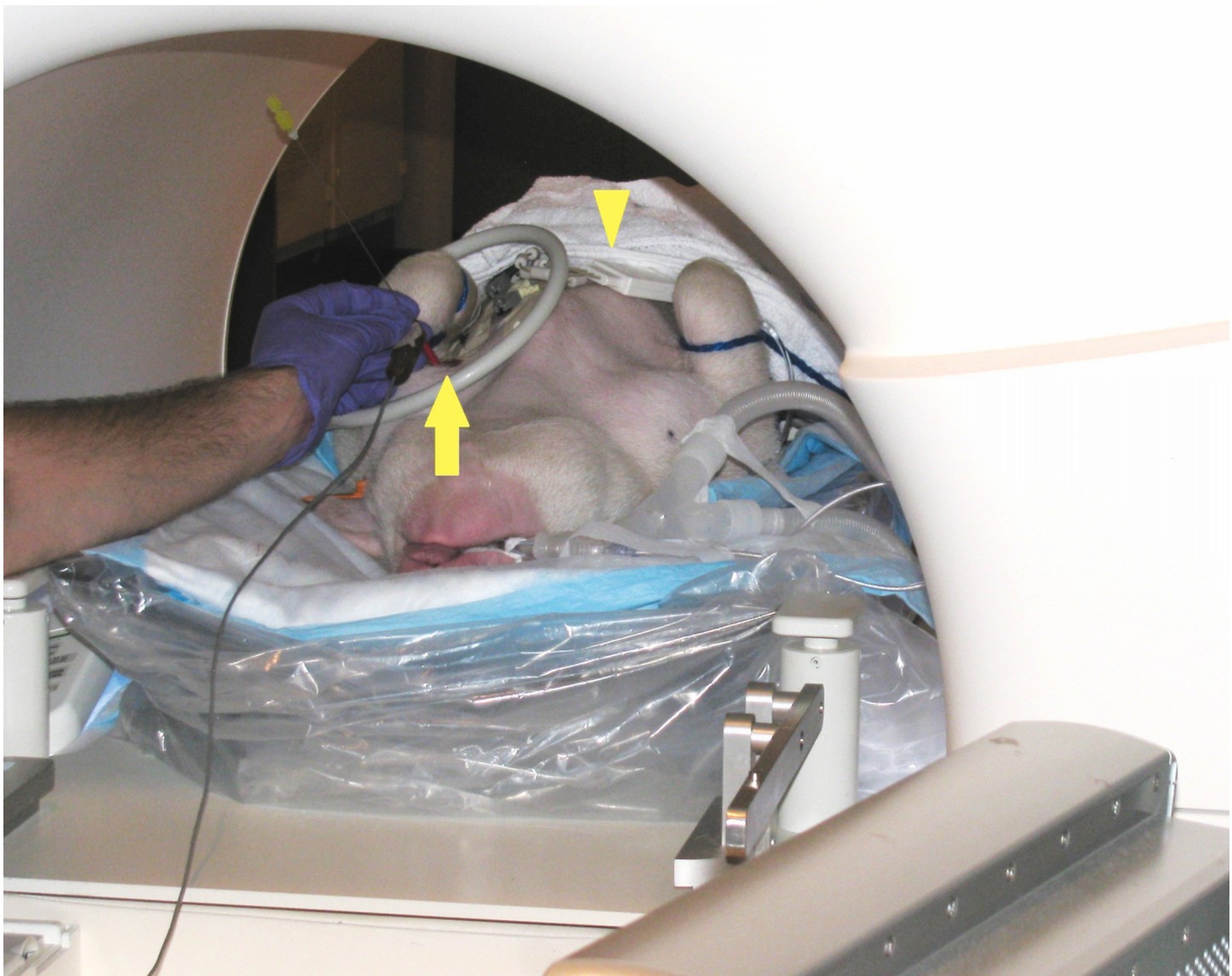

**Fig 6. Setup during biopsy of the shoulder joint in a porcine model.** Navigation probe (arrow) attached to the needle. Flex loop large surface coil (arrowhead) is used for imaging.

## Phantom study

For accuracy evaluation a gel phantom was used. To simulate multiple lesions within the phantom, 15 plastic rings (14 mm inner diameter, 2 mm height) were embedded in the gel at different locations and levels. Fifteen real-time MR-guided punctures aimed at the center of the rings were attempted twice by two interventional radiologists, one with 15 years of clinical practice experience/14 years of iMRI experience (IR1) and one with 2 years of clinical practice experience/1 year of iMRI experience (IR2). Of the first set of 15 punctures (experiment 1), 9 were performed by IR1 and 6 by IR2; for the second set of 15 punctures (experiment 2), 6 were performed by IR1 and 9 by IR2. The in-room monitor was used for interactive navigation with the gradient-based system overlay. On magnified (x3) post-puncture images the distance from the target, defined as the center of the plastic ring, to the needle tip position was measured in 3 dimensions (x, y and z). The

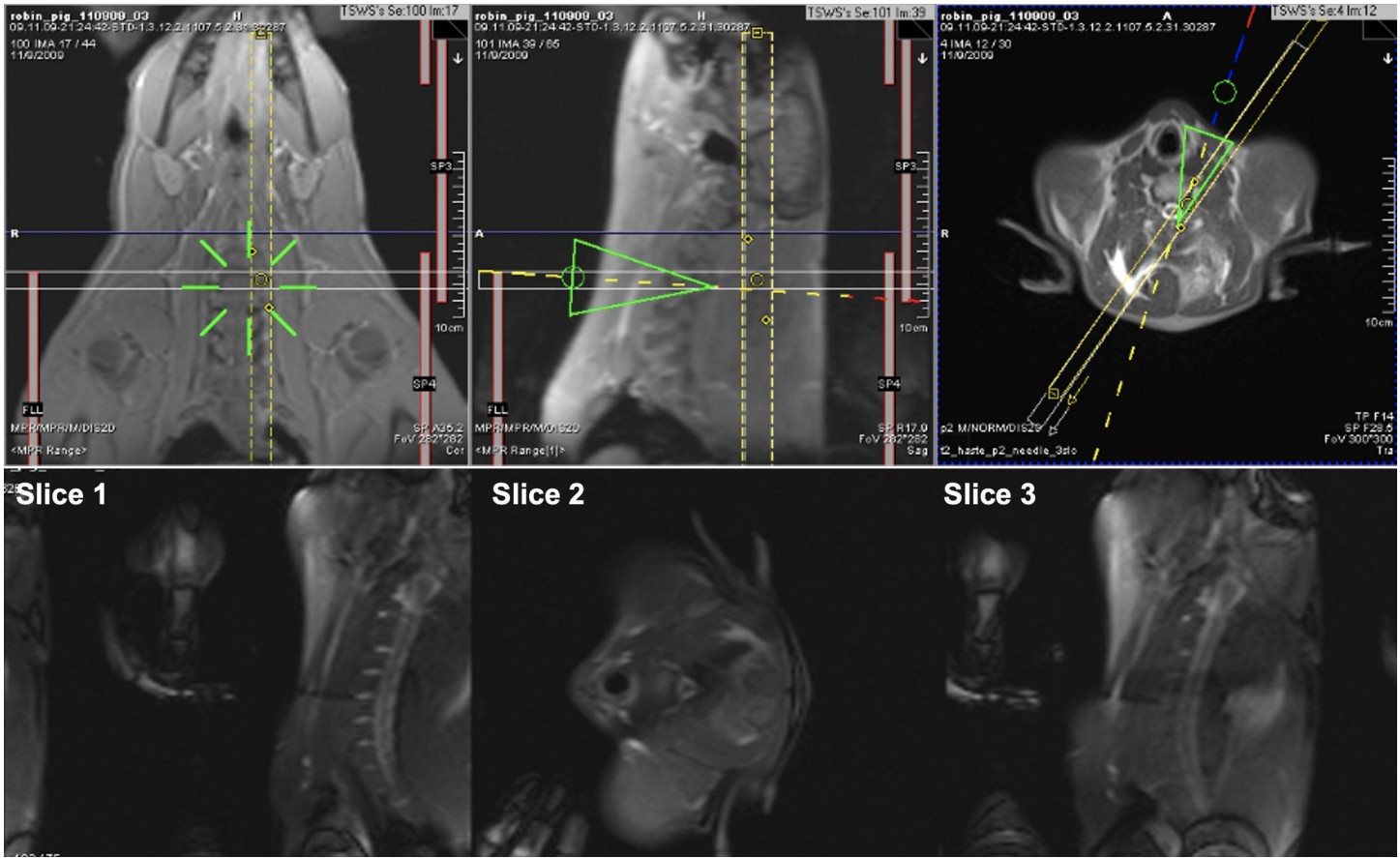

**Fig 7. Cervical spine puncture.** Top Row: The EndoScout graphic overlay, with real-time tracking updates, is superimposed on the three underlying static slices (cor, sag, ax) on the scanner user interface with the tip of the green triangles (middle and right) indicating the needle tip position, and the lines going through the triangles indicating the direction of the needle (yellow: proximal, blue: in-plane, and red: distal relative to the respective slices). Bottom Row: The real-time MR imaging feedback in a separate window (three real-time slices) provides depth information as the needle is advanced and confirms the actual needle position. The three real-time acquired slices are displayed in the inline display window of the scanner user interface in a mosaic mode, with Slice 1 (sag) always following the needle path and Slices 2 (ax) and 3 (sag) updated with real-time images for reference, but without slice location change.

3D distance ($D_{3D}$, overall error), was calculated as: $D_{3D} = \sqrt{\Delta x^2 + \Delta y^2 + \Delta z^2}$, where $\Delta$x, $\Delta$y, and $\Delta$z are the distances from needle tip to target in each x, y, and z direction, respectively.

The 3D overall error and the entry to target time for both radiologists were evaluated for both experiments using an unpaired two-tailed t-test with Welch's correction (significance level $\alpha = 0.05$). A temporal learning curve was estimated by comparing the 3D overall mean error as well as entry to target time of the first half to the second half of the data for each radiologist (unpaired two-tailed t-test with Welch's correction, $\alpha = 0.05$).

## Animal study

A porcine model, approved by the Institutional Animal Care and Use Committee (IACUC Number: JHU IACUC SW 07M390, Maryland, USA), was selected for in vivo evaluation of the feasibility and accuracy of the gradient-based navigation system for MR guided percutaneous interventions. Animals housed at Johns Hopkins University receive species-appropriate environmental enrichment that meets each species needs: Pigs were fed a standard diet and were housed in pairs or social groups, typically by maintaining animals in the social arrangements in which they arrive. Positive human interactions were received during daily husbandry

tasks. Pigs were provided with one hanging toy and at least one floor toy. They receive novel treats during positive interactions with the enrichment staff at least once weekly. Treats may be directly handed to the pigs or distributed in feeder the pigs must use their snouts to root with in order to free the treats.

Four healty pigs (ARCHER FARMS, INC.; Darlington, MD) were used in this study. Pigs were fasted for 24 hours prior to the study. Preoperative sedation was induced with a dose of Ketamine IM (10 mg/kg). After adequate sedation, the animal was intubated with an endotracheal tube. General anaesthesia was initiated with gaseous Isofluorine (2%) and also conducted with Acepromazine (0.5 mg/kg), Atropine (0.8 mg/kg), and Thiopental (15 mg/kg). During the interventions no adverese effects were observerd.

After the experiments, the animal was euthanized with an IV injection of thiopental (100mg/kg) under general anesthesia. Confirmation of death was verified by EKG, cardiac auscultation; as well as cessation of breathing and no reaction to a painful stimulus.

Using 20cm long 20-G MR-compatible metal needles (MReye Chiba Biopsy Needle, Cook Incorporated, Bloomington, IN), six separate punctures were performed by one radiologist (IR1) on four pigs. Targets for percutaneous punctures were a minor calyx in the kidney to simulate a nephrostomy (3), the anterior edge of the glenoid to simulate an anterior approach to shoulder arthrography (1) the cervical intervertebral foramen to simulate a transforaminal epidural injection (2). Time for setup of the EndoScout system was measured starting when MR-images for planning were available and ending with the insertion of the needle. The total needle insertion time started with the skin entry of the needle and ended when the needle reached the target. On magnified (x3) post-puncture images the distance between the target as defined on the planning images and the final needle tip position (overall error) was measured in x, y and z directions. The deviation of the needle tip to the guidance needle trajaectory calculated by the navigation system (system error) was measured.

## Results

### Phantom study

In the phantom study, both radiologists successfully placed the tip of the needle inside the targets in all fifteen 14-mm diameter rings (Fig 5). The overall error (distance from the target, defined as the center of the plastic ring, to the needle tip position) was 7.6±6.0 mm and 5.2±3.1 mm, respectively, for the two times of the puncture experiments. The error in x, y and z directions (relative to the target) was 2.9±2.7, 4.5±6.6, 3.2±2.3 mm respectively for experiment 1 and 2.6±1.7, 2.7±3.8, 2.1±1.2 mm respectively for experiment 2. The less-experienced radiologist pushed the needle too deep inside the ring (overshooting of the needle insertion) in two punctures (one in each experiment) which caused two outlier data points (more than 15 mm error in y direction, perpendicular to the ring). Omitting these two data points, the overall error was 6.2±2.9 mm (experiment 1; 2.9±2.8, 2.9±2.7, 3.1±2.4 mm in x, y and z directions) and 4.4±1.3 mm (experiment 2; 2.6±1.8, 1.8±1.6, 2.2±1.2 in x, y and z directions). The overall error for the two radiologists was 5.4±2.9 mm (9 targets, 3.3±2.9, 2.6±2.5, 2.3±1.5 mm in x, y and z directions) and 7.6±2.6 mm (5 targets, outlier target excluded, 2.3±2.8, 3.6±3.2, 5.0±2.6 mm in x, y and z directions) respectively for experiment 1 and 4.1±1.2 mm (6 targets, 2.1±1.6, 1.6±1.5, 2.3±1.5 mm in x, y and z directions) and 4.7±1.4 mm (8 targets, outlier target excluded, 2.9±1.8, 1.9±1.8, 2.1±1.0 mm in x, y and z directions) respectively for experiment 2. There was no significant difference in the 3D overall error for the two radiologists for experiment 1 (p = 0.19) and 2 (p = 0.40). Furthermore, a learning curve between the experiments 1 and 2 for IR1 (p = 0.24) and IR2 (p = 0.07) could not be detected. The entry to target time was 118.92±91.42 sec for experiment 1 and 137.53±77.77 sec for experiment 2, with 15 targets

each. The two radiologists had an average entry to target time of 57.43±6.05 sec (IR1, 9 targets) and 190.67±92.42 sec (IR2, 6 targets) for the first experiment and 75.50±14.35 sec (IR1, 6 targets) and 178.89±75.14 sec (IR2, 9 targets) for the second experiment. IR1 had a significantly shorter entry to target time compared to IR2 for experiment 1 (p = 0.02) and experiment 2 (p<0.01).

## Animal study

The needle was successfully advanced into all six planned targets. The results for the in-vivo experiment were summarized in Table 1. The overall error (error in placement between the target and the final needle tip position) and the system error (deviation of the needle tip from the guidance needle trajectory overlay calculated by the navigation system) in six punctures was 8.8±7.8 mm (min: 0.8 mm; max: 20.0 mm) and 3.3±1.4 mm (min: 1.8 mm; max: 5.2 mm), respectively. The total needle insertion time (entry point to target time) for all planned target was 96.3±40.5 seconds (min: 42 seconds; max: 154 seconds). Similar to the phantom experiments, in two punctures the needle was advanced too far by IR1.

## Discussion

In iMRI, navigation systems for percutaneous needle placements (e.g. biopsies or ablations) should interactively provide needle pathway planning, entry point and target selection, without restricting the freedom of navigation in the unlimited trajectories unique to MRI. Additional real time imaging feedback during the needle placement should be provided to allow the operator to adjust for intraprocedural anatomic distortion and needle deviation, in order to increase the safety and precision of the needle placement [10, 13, 29, 30]. The gradient-based navigation system combined with the interactive real-time sequence evaluated in this study facilitates interactive procedures in a wide-bore MR system. The integrated system was able to reliably navigate the needle and superimpose the real-time updated needle trajectory on pre-acquired images. This facilitated needle insertion along the planned paths and allowed for confirmation of successful targeting. The multi-slice real-time images were acquired and displayed in a mosaic mode to enable device guidance while allowing the operator to adapt to both organ movement due to breathing and tissue deformation/displacement due to needle insertion. This is superior to other approaches that use out-of-bore navigation [15, 16, 24, 31] without real-time feedback, and therefore do not allow the operator to adapt to target motion. This is especially problematic when targeting lesions in soft, mobile organs such as the liver that suffer from both respiratory motion and tissue deformation. Nevertheless, comparing the mean total error in all directions to other publications (mean error of 0.99mm±0.47mm to 3.9mm ±2.4mm) the presented gradient based technique shows a slightly lower accuracy [11, 13, 15, 32, 33]. This is compensated for because the integrated system uses MRI gradients rather than

**Table 1. Summary of porcine punctures.**

| Target # | Pig # | Puncture Target | Entry to Target Time (seconds) | Overall Error (mm) | System Error (mm) |
|---|---|---|---|---|---|
| 1 | 1 | Shoulder Joint | 108 | 20.0 | 2.9 |
| 2 | 1 | Cervical spine 1 | 72 | 4.6 | 1.8 |
| 3 | 1 | Cervical spine 2 | 154 | 0.8 | 1.9 |
| 4 | 2 | Kidney | 42 | 6.9 | 5.2 |
| 5 | 3 | Kidney | 125 | 3.4 | 3.4 |
| 6 | 4 | Kidney | 79 | 17.0 | 4.8 |
| All | | Mean ± STD | 96.7 ± 40.3 | 8.8 ± 7.8 | 3.3 ± 1.4 |

optical tracking, thus eliminating any line of sight problems that could arise from in-bore needle manipulation, where the field of view of cameras is restricted. In addition, the tracking tool utilizes the gradient system of the MR scanner rather than cameras, optical markers, robotic arms, etc. Therefore, the system is easy to implement and readily usable with short setup time.

There are other types of navigation techniques for MRI guided punctures. MRI compatible robotic devices [14, 19–21] usually require extensive setup time, provide reliable guidance, but due to medicolegal reasons a physician is still required to advance the needle. Other systems use augmented reality to project information acquired inside the magnet onto the patient for puncture guidance [15, 16, 22]. Although relatively intuitive, these systems lack real-time feedback for adjustments of needle and therefore do not allow the operator to adapt to motion/deformation. A good example is simple grid-based Cartesian devices, which are used only for breast and prostate biopsies where motion is not a concern [34, 35]. However, in contrast to most robotic and augmented reality systems, these grid based systems are commercially available and are used extensively.

Freehand punctures without navigation tools can also be performed in MR systems using for instance standard radiological markers or body landmarks. This approach has no regulatory restrictions but has been shown to be relatively time consuming due to what can be a cumbersome preparation period [36]. There are reports suggesting that efficiency of both access and freehand punctures is improved in "vertically-open" MR systems which facilitate lateral access to the patient [12] as opposed to "open-bore" systems that only allow patient access from the ends of the magnet. However, most "vertically-open" MR systems operating at 1T or above are no-longer being produced or supported by the manufacturers. Currently available "vertically-open" magnets operate at low field strengths and therefore produce suboptimal image quality compared to "open-bore" systems especially when real-time imaging is needed.

In contrast, the method presented here allows for high-quality real-time image collection before and during percutaneous needle placement that can be used as overlay during the procedure. Ultimately, we envision a system that can use real-time information for the adjustment of the pre-acquired images. This would reduce the complexity of the information and increase the ease of use of the system. Along the same line, needle bending might also add to the complexity of the procedure as the trajectory and the real needle do not align. Adding two real-time slices parallel to the central slice as described with freehand punctures [11] might be helpful in these situations. Another solution could be an automatic adjustment of the trajectory and the needle path based on real-time information. However, this would require recognition of the inserted needle which is not available, yet. But even without these additional features, the system's gradient-based tracking simplifies the initial stages of the intervention, when entry point and trajectory are being determined while most of the needle is outside of the patient and still surrounded by air. In this phase of the intervention, real-time imaging is not helpful since the needle artifact itself is not visible in air.

There are limitations with this study. First, the number of animal experiments is relatively small and no control group (e.g. with freehand punctures) was included. Future work could include a two-armed clinical trial with and without the gradient-based navigation tool as an add-on. Second, we targeted only organs that have minimal motion during breathing, due to the fact that the graphic overlay is not registered to breathing. Since the pre-acquired MR images do not adjust for breathing motion, real-time images were acquired during the puncture for identifying moving targets as well as confirming the needle position. The combination of static planning and real-time imaging is a difficult task for the interventionalists as they have to follow complex information during the puncture. Third, we observed overshooting of the needle in both, phantom and animal experiments. This is understandable since the navigation system does not track the actual needle tip and the interventionalist relies heavily on the

real-time image feedback, which has a frame rate significantly slower than that of the tracking device. In addition, going beyond the planned target with the needle tip was often performed on purpose to overcome the elasticity of the target tissue, especially during phantom and animal tests when we use needles that may have been used many times before and are less sharp. Given the precise placement in the other two directions, we don't expect that this to be an issue in clinical practice when new and sharp needles are used.

In summary, we integrated a gradient-based navigation system with an interactive real-time sequence for improved MRI guidance of percutaneous punctures. The integrated system was capable of reliably navigating a needle, superimposing both the device's position and orientation in real-time on pre-acquired images and displaying needle and target in real-time. Based upon the results of this pilot study, we conclude that MR image guidance of percutaneous punctures with the EndoScout interactive navigation system as an add-on is feasible and accurate. The system provided most intuitive help during the selection of the entry point and orientation of the needle in a wide-bore MR system. Usability seems to be independent from iMRI expertise since both interventional radiologists produced similar results and did not experience a learning curve. Our results on accuracy in phantoms and technical feasibility in animals suggest that the Endoscout System can be implemented in the clinical realm. However, the current prototype is not ready for clinical use. Next steps include registration to breathing motion and integration of Endoscout with new interfaces such as Access-I (Siemens Healthineers, Germany, Erlangen). We expect that this will facilitate clinical implementation of the tracking system, which could be especially helpful for more complex procedures that require double oblique puncture trajectories or positioning of multiple needles or applicators targeting several lesions, such as in thermal ablations.

## Supporting information

**S1 Checklist. The ARRIVE guidelines checklist.**
(PDF)

## Author Contributions

**Conceptualization:** Li Pan, Steffi Valdeig, Kun Qing, Barry Fetics, Amir Roth, Erez Nevo, Clifford R. Weiss, Frank K. Wacker.

**Data curation:** Li Pan, Kun Qing, Clifford R. Weiss, Frank K. Wacker.

**Funding acquisition:** Li Pan.

**Investigation:** Li Pan, Steffi Valdeig, Kun Qing, Barry Fetics, Amir Roth, Erez Nevo, Clifford R. Weiss, Frank K. Wacker.

**Methodology:** Li Pan, Steffi Valdeig, Kun Qing, Barry Fetics, Amir Roth, Erez Nevo, Clifford R. Weiss, Frank K. Wacker.

**Resources:** Barry Fetics, Amir Roth, Erez Nevo.

**Software:** Li Pan, Barry Fetics, Amir Roth, Erez Nevo.

**Supervision:** Frank K. Wacker.

**Validation:** Li Pan, Steffi Valdeig, Kun Qing, Frank K. Wacker.

**Visualization:** Li Pan, Frank K. Wacker.

**Writing – original draft:** Li Pan, Frank K. Wacker.

**Writing – review & editing:** Steffi Valdeig, Urte Kägebein, Kun Qing, Barry Fetics, Amir Roth, Erez Nevo, Bennet Hensen, Clifford R. Weiss, Frank K. Wacker.

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
