## [Decision Letter · Decision Letter 0]

26 May 2020

PONE-D-20-07053

Integration and Evaluation of a Gradient-based Needle Navigation System for Percutaneous MR-guided Interventions

PLOS ONE

Dear Dr. Hensen,

Thank you for submitting your manuscript to PLOS ONE. After careful consideration, we feel that it has merit but does not fully meet PLOS ONE’s publication criteria as it currently stands. Therefore, we invite you to submit a revised version of the manuscript that addresses the points raised during the review process.

We look forward to receiving your revised manuscript.

Kind regards,

Zhongliang Zu, Ph.D.

Academic Editor

PLOS ONE

2. Please include the following information in your methods section:

•        Please specify the source from which animals were obtained

•        Please provide details of animal welfare and housing (e.g., shelter, food, water, environmental enrichment)

•        Please complete and submit a copy of the ARRIVE Guidelines checklist, a document that aims to improve experimental reporting and reproducibility of animal studies for purposes of post-publication data analysis and reproducibility: https://www.nc3rs.org.uk/arrive-guidelines. Please include your completed checklist as a Supporting Information file. Note that if your paper is accepted for publication, this checklist will be published as part of your article.

"I have read the journal's policy and the authors of this manuscript have the following competing interests:Li Pan = full employee of Siemens Healthineers; Kun Qing = receives grant support from Siemens Healthineers and was an intern employee while working on this Project; Barry Fetics/Amir Roth = employee of Robin Medical; Erez Nevo = founder and stock owner in Robin Medical; Clifford Weiss = receives grant support from Siemens Healthineers, for BTG and Medtronic, is a consultant for BTG and Medtronic; Frank Wacker = receives institutional funding from Siemens"

We note that one or more of the authors are employed by a commercial company: Siemens Healthineers, Robin Medical, BTG and Medtronic

5. Please upload a new copy of Figure 4 as the detail is not clear. Please follow the link for more information: https://blogs.plos.org/plos/2019/06/looking-good-tips-for-creating-your-plos-figures-graphics/

Reviewers' comments:

Reviewer's Responses to Questions

**Comments to the Author**

1. Is the manuscript technically sound, and do the data support the conclusions?

Reviewer #1: Yes

Reviewer #2: Partly

Reviewer #3: Yes

2. Has the statistical analysis been performed appropriately and rigorously? 

Reviewer #1: N/A

Reviewer #2: Yes

Reviewer #3: Yes

3. Have the authors made all data underlying the findings in their manuscript fully available?

Reviewer #1: Yes

Reviewer #2: Yes

Reviewer #3: Yes

4. Is the manuscript presented in an intelligible fashion and written in standard English?

Reviewer #1: Yes

Reviewer #2: Yes

Reviewer #3: Yes

5. Review Comments to the Author

Reviewer #1: This is an interesting paper. The following points should be considered to further improve it.

1. Figures 1, 3, 4 are very low resolution. Please replace with high DPI version.

2. The results describe the performance of the proposed system but no control group was presented. The Authors should at least provide a comparison based on the literature (e.g. needle insertion time, error in plane and in 3D for competitor solutions).

3. The Authirs write on Page 21 "The multi-slice real-time images were acquired and displayed in a mosaic mode to enable device guidance while allowing the operator to adapt to both organ movement due to breathing and tissue deformation/displacement due to needle insertion."

Later however the Authors write (on Page 23) "we targeted only organs that have minimal motion during breathing". Please explain the reason for this choice, since the system should be able to cope with this, right?

4. On page 23 the Authors write "we observed overshooting of the needle in both, phantom and animal experiments". A relatively simple solution to this problem would be a depth gauge on the needle. Is this not possible with the proposed system?

Reviewer #2: The manuscript is interesting and well written. I can see the interest of the system but do the results really support that this helps the IR? not sure (8mm deviation, 4 cases of overshooting)... it is not an issue as it is work in progress, but this should be mentioned in the discussion especially Given the conflict of interests of the authors,

A couple of points that need to be addressed/corrected to my point of view:

- is the needle holder sterilizable?

- please provide the characteristics of the BEAT-IRTTT sequences

- line 168: in which plane were HASTE images acquired? was was the number of HASTE images?

- fig7 is mentioned before fig6 in the text

- line 223: what were the reasons to use a T1 or a T2 sequence? please be more precise

- what were the targets in the pigs? Kidney is rather vague. Same for cervical spine or shoulder. a figure could be helpful

- results section and measures: given the low spatial resolution of MRI guidance, giving a hundredth of millimeter does not make sense. there is a huge place for errors there. Who made the measurements? how many times were they made?

- line 282 to 295: you give a lot of measures but the terms are unclear. It would be better to specify clearly which results apply to expriement1/2 and IR1/2

- was overshooting recognized at the time of puncture? if not, i believe that statistic should be presented including the outlier target.

- line 291: learning curve: how do you compare the learning curve? it was not specified in the method section. Moreover, base on the length of procedure, it seems to be a difference

- please provide a p-value if there is one to compare precision and time to insert the needle between IR1 and IR2

- line 299 to 305: these are not results. It is a subjective description, please remove

Reviewer #3: This study aimed to investigate the feasibility, accuracy and clinical implementation with use of an animal model of a grading-based navigation system for MR guided interventions using an open bore 1.5 Tesla MRI system. This is a timely topic as interventional MRI procedures gain traction and thus navigation systems are important and will become even more important in the future. The navigation system is well described. The methods and statistical analysis are described in detail and to my review appropriate. The manuscript is very well written, without major requirements for edits. The figures and illustrations are excellent. The manuscript includes detailed information for readers to implement and reproduce the techniques that result of this study, respectively. The author group is well known to the field of interventional magnetic resonance imaging including several pioneers and clinically active interventional radiologists, which carries a high credibility for the study.

6. PLOS authors have the option to publish the peer review history of their article (what does this mean?). If published, this will include your full peer review and any attached files.

Reviewer #1: No

Reviewer #2: No

Reviewer #3: No

---

## [Author Response · Author response to Decision Letter 0]

16 Jun 2020

Dear Dr. Zu,

I would like to thank you and your reviewers for the time and effort you have put into the review of our manuscript. We really appreciate all of your input. The following is a detailed summary of the replies and changes made in our manuscript in direct response to each specific reviewer comment.

Sincerely, 

Bennet Hensen and Frank Wacker, for all authors

The manuscript style was adapted to the style requirements.

2. Please include the following information in your methods section:

• Please specify the source from which animals were obtained

• Please provide details of animal welfare and housing (e.g., shelter, food, water, environmental enrichment)

• Please complete and submit a copy of the ARRIVE Guidelines checklist, a document that aims to improve experimental reporting and reproducibility of animal studies for purposes of post-publication data analysis and reproducibility: https://www.nc3rs.org.uk/arrive-guidelines. Please include your completed checklist as a Supporting Information file. Note that if your paper is accepted for publication, this checklist will be published as part of your article.

Detailed information on animal source and animal welfare and housing was added. 

The ARRIVE Guidelines checklist was uploaded. 

(…)

An updated Funding Statement and a Competing Interests Statement was included in the cover letter. 

4. PLOS requires an ORCID iD for the corresponding author in Editorial Manager on papers submitted after December 6th, 2016. 

The ORCID iD was added. 

5. Please upload a new copy of Figure 4 as the detail is not clear. Please follow the link for more information: https://blogs.plos.org/plos/2019/06/looking-good-tips-for-creating-your-plos-figures-graphics/

All Figures are now replaced with high DPI versions. Figure 4 has been adjusted and homogenized, the sequence diagrams now look the same.

Review Comments to the Author:

Reviewer #1: This is an interesting paper. The following points should be considered to further improve it.

1. Figures 1, 3, 4 are very low resolution. Please replace with high DPI version.

All Figures are now replaced with high DPI versions.

2. The results describe the performance of the proposed system, but no control group was presented. The Authors should at least provide a comparison based on the literature (e.g. needle insertion time, error in plane and in 3D for competitor solutions).

We included a comparison based on the literature in the “Discussion” section (line 533 – 536).

3. The Authors write on Page 21 "The multi-slice real-time images were acquired and displayed in a mosaic mode to enable device guidance while allowing the operator to adapt to both organ movement due to breathing and tissue deformation/displacement due to needle insertion."

Later however the Authors write (on Page 23) "we targeted only organs that have minimal motion during breathing". Please explain the reason for this choice, since the system should be able to cope with this, right?

The current system includes static as well as real time information. This is now explained in the “Discussion” section (line 587 – 592).

4. On page 23 the Authors write "we observed overshooting of the needle in both, phantom and animal experiments". A relatively simple solution to this problem would be a depth gauge on the needle. Is this not possible with the proposed system?

We agree with the reviewer that a depth gauge could be a simple solution. However, elasticity of soft tissue in humans, animals and most phantoms cause pushback once the target is reached. In order to adjust for elastic resistance, overshooting is somewhat intended but the extent is sometimes difficult to foresee, especially when relying on needle tip visualization in static rather than realtime images. 

Reviewer #2: The manuscript is interesting and well written. I can see the interest of the system but do the results really support that this helps the IR? not sure (8mm deviation, 4 cases of overshooting)... it is not an issue as it is work in progress, but this should be mentioned in the discussion especially Given the conflict of interests of the authors,

We agree with the reviewer. This is a prototype system not ready for clinical use. We clarified this in the “Discussion” section (line 613 – 617).

A couple of points that need to be addressed/corrected to my point of view:

- is the needle holder sterilizable?

Yes, this information were added in the manuscript (line 113)

- please provide the characteristics of the BEAT-IRTTT sequences

Details for BEAT-IRTTT were added in the methods section (line 161 - 164)

- line 168: in which plane were HASTE images acquired? What was the number of HASTE images?

The HASTE sequence was acquired in an axial, coronal and transversal plane (line 187).

- fig7 is mentioned before fig6 in the text

This was corrected.

- line 223: what were the reasons to use a T1 or a T2 sequence? please be more precise

T1 images were selected. This information has been corrected in the manuscript.

- what were the targets in the pigs? Kidney is rather vague. Same for cervical spine or shoulder. a figure could be helpful

Spine: the cervical intervertebral foramen to simulate a transforaminal epidural injection

Shoulder joint: the anterior edge of the glenoid to simulate an anterior approach to shoulder arthrography

Kidney: minor calyx of the lower pole to simulate a nephrostomy

Details are now given in the manuscript (line 298-300)

- results section and measures: given the low spatial resolution of MRI guidance, giving a hundredth of millimeter does not make sense. there is a huge place for errors there. Who made the measurements? how many times were they made?

We fully agree with the reviewer, that this does not make sense. Therefore, the numbers were rounded. 

- line 282 to 295: you give a lot of measures but the terms are unclear. It would be better to specify clearly which results apply to expriement1/2 and IR1/2

In the “Results” section, the measurements are given for experiment 1 and 2 and for IR 1 and 2 respectively. We clarified this by adding some information regarding IR 1 and 2. 

- was overshooting recognized at the time of puncture? if not, i believe that statistic should be presented including the outlier target.

Overshooting was recognized during the intervention by the less experienced radiologist who overestimated elasticity and prematurely finished the respective puncture. For full disclosure the experiments are mentioned but, give the operator error, are not included in the results.

- line 291: learning curve: how do you compare the learning curve? it was not specified in the method section. Moreover, base on the length of procedure, it seems to be a difference

An operator not familiar with a guidance system usually improves after the first set of experiments. In order to account for a potential learning curve, we therefore compared experiment 1 and 2 for both IR1 and 2. This was clarified in the “Results” section. 

- please provide a p-value if there is one to compare precision and time to insert the needle between IR1 and IR2

This information is given in the “Results” section (time to insert: line 463; precision: line 456).

- line 299 to 305: these are not results. It is a subjective description, please remove

We agree. The paragraph is omitted.

---

## [Decision Letter · Decision Letter 1]

6 Jul 2020

Integration and Evaluation of a Gradient-based Needle Navigation System for Percutaneous MR-guided Interventions

PONE-D-20-07053R1

Dear Dr. Hensen,

We’re pleased to inform you that your manuscript has been judged scientifically suitable for publication and will be formally accepted for publication once it meets all outstanding technical requirements.

Kind regards,

Zhongliang Zu, Ph.D.

Academic Editor

PLOS ONE

Additional Editor Comments (optional):

Reviewers' comments:

Reviewer's Responses to Questions

**Comments to the Author**

1. If the authors have adequately addressed your comments raised in a previous round of review and you feel that this manuscript is now acceptable for publication, you may indicate that here to bypass the “Comments to the Author” section, enter your conflict of interest statement in the “Confidential to Editor” section, and submit your "Accept" recommendation.

Reviewer #1: All comments have been addressed

2. Is the manuscript technically sound, and do the data support the conclusions?

Reviewer #1: Yes

3. Has the statistical analysis been performed appropriately and rigorously? 

Reviewer #1: Yes

4. Have the authors made all data underlying the findings in their manuscript fully available?

Reviewer #1: Yes

5. Is the manuscript presented in an intelligible fashion and written in standard English?

Reviewer #1: Yes

6. Review Comments to the Author

Reviewer #1: The Authors have addressed all comments from this reviewer. In my opinion, the manuscript is ready for publication.

7. PLOS authors have the option to publish the peer review history of their article (what does this mean?). If published, this will include your full peer review and any attached files.

Reviewer #1: No

---

## [Editor Report · Acceptance letter]

8 Jul 2020

PONE-D-20-07053R1 

Integration and Evaluation of a Gradient-based Needle Navigation System for Percutaneous MR-guided Interventions 

Dear Dr. Hensen:

I'm pleased to inform you that your manuscript has been deemed suitable for publication in PLOS ONE. Congratulations! Your manuscript is now with our production department. 

Kind regards, 

on behalf of

Dr. Zhongliang Zu 

Academic Editor

PLOS ONE